# Stable Carbon and Nitrogen Isotope Signatures in Three Pondweed Species—A Case Study of Rivers and Lakes in Northern Poland

**DOI:** 10.3390/plants14152261

**Published:** 2025-07-22

**Authors:** Zofia Wrosz, Krzysztof Banaś, Marek Merdalski, Eugeniusz Pronin

**Affiliations:** Department of Plant Ecology, Faculty of Biology, University of Gdansk, 80-309 Gdansk, Poland; z.wrosz.167@studms.ug.edu.pl (Z.W.); krzysztof.banas@ug.edu.pl (K.B.); marek.merdalski@ug.edu.pl (M.M.)

**Keywords:** *δ*^13^C, *δ*^15^N, macrophytes, organic matter, water conditions, lakes, rivers

## Abstract

Aquatic plants, as sedentary lifestyle organisms that accumulate chemical substances from their surroundings, can serve as valuable indicators of long-term anthropogenic pressure. In Poland, water monitoring is limited both spatially and temporally, which hampers a comprehensive assessment of water quality. Since the implementation of the Water Framework Directive (WFD), biotic elements, including macrophytes, have played an increasingly important role in water monitoring. Moreover, running waters, due to their dynamic nature, are susceptible to episodic pollution inputs that may be difficult to detect during isolated, point-in-time sampling campaigns. The analysis of stable carbon (*δ*^13^C) and nitrogen (*δ*^15^N) isotope signatures in macrophytes enables the identification of elemental sources, including potential pollutants. Research conducted between 2008 and 2011 encompassed 38 sites along 15 rivers and 108 sites across 21 lakes in northern Poland. This study focused on the isotope signatures of three pondweed species: *Stuckenia pectinata*, *Potamogeton perfoliatus*, and *Potamogeton crispus*. The results revealed statistically significant differences in the *δ*^13^C and *δ*^15^N values of plant organic matter between river and lake environments. Higher *δ*^15^N values were observed in rivers, whereas higher *δ*^13^C values were recorded in lakes. Spearman correlation analysis showed a negative relationship between *δ*^13^C and *δ*^15^N, as well as correlations between *δ*^15^N and the concentrations of Ca^2+^ and HCO_3_^−^. A positive correlation was also found between *δ*^13^C and dissolved oxygen levels. These findings confirm the utility of *δ*^13^C and, in particular, *δ*^15^N as indicators of anthropogenic eutrophication, including potentially domestic sewage input and its impact on aquatic ecosystems.

## 1. Introduction

Macrophytes play a key role in the functioning of lake and river ecosystems. They constitute an important structural component of these above-mentioned environments, influencing the biogeochemical cycling of elements, sediment stabilization, and habitat conditions for aquatic organisms [1]. Due to their persistence and sensitivity to changes in the physicochemical parameters of water, macrophytes, along with other biotic elements such as phytoplankton, phytobenthos, macrozoobenthos, and ichthyofauna, are among the key components used to assess the ecological status of rivers and lakes [2]. Species composition, abundance, spatial coverage, and physiological condition give macrophytes the status of bioindicators of the aquatic environment [3,4]. The use of aquatic plants makes it possible to identify trophic changes in both lentic and lotic ecosystems [5,6,7]. Since 2000, the use of macrophytes in the ecological assessment of water bodies has been mandated by the European Union’s (EU) Water Framework Directive (WFD), which, in Article 8, requires the monitoring of water status based on biological elements, including aquatic vegetation [8].

It is precisely characteristics such as a short life cycle, high production rates, and sensitivity to changes in water chemistry that make macrophytes effective early warning indicators of ongoing changes in aquatic ecosystems and the influx of pollutants [3,9,10,11]. Research has shown that isotope signatures can support water quality monitoring efforts [12]. Some macrophyte species do not respond immediately to changes in water chemistry; however, they retain a record of spontaneous, irregular pollution inputs in their isotope signatures [13]. The carbon and nitrogen isotope signatures in macrophytes provide information about the sources of these elements within the plant [14]. Isotopic composition analyses can provide more precise information on the input of allochthonous matter from the catchment area and its relationship to land use patterns within that area [13]. By observing subtle changes in water chemistry based on the isotopic signals recorded in macrophytes, it becomes possible to take appropriate action and implement measures aimed at improving water quality.

The greatest challenges facing water management in Poland are the poor quality of surface waters and extensive hydromorphological alterations [15]. Hydrotechnical solutions aimed at reducing flood risk and impact often involve riverbed modifications, including deepening of riverbeds. Such interventions promote the transformation and intensive development of river catchments, subjecting these ecosystems to high levels of anthropogenic pressure [15]. Some surface waters lack adequate buffer zones, making them more vulnerable to accelerated eutrophication [15]. The increase in water trophic status, driven by nutrient inputs from both point and diffuse sources, is the main threat to achieving good ecological status in standing waters in Poland [16]. The agricultural sector is considered the primary source of nutrient pollution [15]. Similar problems are faced by other European countries, including Belarus, Germany, Sweden, Belgium, the Netherlands, Denmark, Ukraine, and Estonia [15,17]. It is worth noting that in Poland, most surface waters are classified as below good in terms of ecological status or potential and as poor when considering the overall status (i.e., ecological and chemical status combined) [15,18]. These insufficient conditions are largely due to elevated eutrophication, which adversely affects the biological parameters on which the ecological status under the WFD is based [19]. Therefore, in the European Union’s water policy, eutrophication is a frequently addressed issue. Directives such as Urban Waste Water Treatment Directive 91/271/EWG, Nitrates Directive 91/676/EWG, and WFD 2000/60/WE contain requirements for the assessment of this phenomenon [16].

Stable isotope signature values (*δ*) allow for the identification of the sources of chemical compounds containing the elements being analyzed [20,21]. In terms of stable carbon and nitrogen isotope composition (*δ*^13^C and *δ*^15^N), general knowledge of macrophyte biochemistry remains insufficiently established [22]. Research on the stable isotope composition of carbon and nitrogen in plants enables inferences about the preferred sources of these elements by aquatic plants, as well as the factors influencing their uptake and selection processes [23,24]. Some of the main factors determining carbon availability and the selection of its source by submerged aquatic plants are the chemical composition and physical properties of the water and sediments [12].

The study of *δ*^13^C allows for an indirect conclusion about the productivity of an aquatic ecosystem [25]. In turn, the variability of *δ*^15^N values in freshwater macrophytes enables the identification of potential sources of pollution inputs, their intensity, and their origin. This is due to differences in the *δ*^15^N signatures of the primary nitrogen sources being utilized [21]. Thus, *δ*^15^N values in macrophytes can be used to infer whether the assimilated nitrogen originated from sources such as agricultural fertilizers or municipal wastewater, as well as to assess their ecological impact on riverine ecosystems [23]. Surface water pollution resulting from agricultural irrigation and wastewater discharge takes place when irrigation of agricultural lands with effluents leads to the transfer of nutrients such as nitrogen and phosphorus into nearby water bodies. Changes in macrophyte species composition in response to elevated nutrient input develop over a longer timescale compared to phytoplankton. Therefore, tracking nutrient influx from diverse sources and understanding their movement through the environment is essential for maintaining the stability of freshwater ecosystems [21]. In addition, this phenomenon can disrupt the competitive balance between phytoplankton and macrophyte species, which in turn may lead to a loss of biodiversity within a given river or lake ecosystem.

Using stable isotope analysis of *δ*^13^C and *δ*^15^N in macrophytes as a tool, a comparison was undertaken of the values obtained from three pondweed species (*Potamogeton perfoliatus* L., *Stuckenia pectinata* (L.) Börner, and *Potamogeton crispus* L.) collected from rivers and lakes in northern Poland. The selected species from the Potamogetonaceae family were chosen due to their differing habitat requirements, their representative presence in *Potamogeton*-type lakes [26,27], and their frequent occurrence in lowland rivers of Poland [28]. The significant difference in water retention time between lakes and rivers (lentic and lotic ecosystems), combined with the continuous replacement and partial mixing of water in rivers, underscores that rivers and lakes differ substantially in their susceptibility to pollution and the intensity of environmental pressures they experience [29]. In flowing waters, once pollutant inputs stop, the ecosystem has the capacity to recover through natural self-purification processes [30,31], and clearer, less polluted water may subsequently appear. In lakes, the chemical composition of the water tends to remain relatively stable, and the supply of substrates required for photosynthesis is more evenly maintained compared to river systems [29]. Water movement in lakes is primarily regulated by temperature fluctuations, resulting in low hydrodynamic activity. Under such conditions, pollutants tend to accumulate, often becoming concentrated in bottom sediments [29,32]. To sum up, rivers typically exhibit lower and more variable nutrient concentrations due to continuous flow and short water residence times, which limit nutrient accumulation. In contrast, lakes, with longer residence times and stratification, allow nutrients to accumulate in the water column and sediments, making them more prone to eutrophication [29]. It is worth pointing out that the literature reports about the isotopic signal of submerged macrophytes in Poland [12,13,20,33,34,35,36] and Europe are very limited [21,37,38,39,40,41,42,43] considering two different types of surface water habitats [42]. Among the studies reviewed, only the work by Osmond et al. [42] compares several pondweed species across different habitats, including rivers and lakes. Their results revealed substantial differences in the *δ*^13^C isotopic signatures, with more negative values observed in riverine habitats and higher (less negative) values in lakes. However, it should be noted that the authors of that study focused exclusively on *δ*^13^C, without addressing other isotopic parameters.

The aim of this study is to assess the variability of *δ*^13^C and *δ*^15^N values in three species of aquatic plants, taking into account their sampling locations—both rivers and lakes. Based on a review of the literature [40,42] and preliminary observations [13,36,38], research hypotheses were formulated assuming that *δ*^13^C and *δ*^15^N values in the studied pondweed species will show significant differences between river and lake ecosystems due to contrasting hydrological conditions, nutrient availability, and water flow characteristics in these environments. It was also assumed that the *δ*^13^C and *δ*^15^N isotope signatures in macrophyte tissues would reflect the influence of anthropogenic pollution sources, with the extent of this influence varying depending on the type of aquatic ecosystem. It was also hypothesized that statistically significant correlations exist between the isotope signatures in the analyzed plants and selected physicochemical parameters of the water. This could enable the identification of key factors driving isotopic fractionation in the studied macrophytes. To the best of our knowledge, this study is the first in Poland, and among the first in Europe, to compare δ^13^C and especially *δ*^15^N signatures of *S. pectinata*, *P. perfoliatus*, and *P. crispus* between river and lake environments. The results highlight clear habitat-related isotopic differences and demonstrate the potential of these species as sensitive bioindicators of eutrophication and nutrient pollution. These findings advance our understanding of the factors shaping isotopic variation in macrophytes and support their use in developing more effective approaches to monitoring aquatic ecosystem health.

## 2. Results

### 2.1. δ^13^C_ORG_ and δ^15^N_ORG_ Values of Investigated Plants in Different Surface Water Habitats

In lakes, a significant difference in *δ*^13^C was observed among all collected specimens (Figure 1). Significantly higher *δ*^13^C values (*p* < 0.05, Dunn’s test following Kruskal–Wallis test) were observed for *S. pectinata* compared to *P. crispus*, while significantly lower *δ*^13^C values were recorded for *P. perfoliatus* compared to the other two species (*p* < 0.05, Dunn’s test following Kruskal–Wallis test). In the case of rivers, statistically significant differences in *δ*^13^C were found only between *P. crispus* and *P. perfoliatus,* where the recorded values of *P. perfoliatus* were significantly higher (Figure 1, Dunn’s test *p* < 0.05 test following Kruskal–Wallis test).

The *δ*^15^N values in macrophytes collected from lakes and rivers varied among species in a similar pattern (Figure 2). For all investigated species, the *δ*^15^N values were higher in the river than in the lakes, and those differences were statistically significant (Mann–Whitney U test; *p* < 0.05). Comparing values recorded in lakes, we found only significant differences between *P. crispus* and *P. perfoliatus,* which have the highest recorded values of *δ*^15^N in investigated lakes (*p* < 0.05, Dunn’s test following Kruskal–Wallis test). In contrast, no statistically significant differences (*p* > 0.05, Dunn’s test following Kruskal–Wallis test) in *δ*^15^N were found in the organic matter of macrophytes from rivers (Figure 2).

Additionally, the lowest *δ*^13^C and *δ*^15^N values were observed in *S. pectinata* individuals collected from flowing waters. Furthermore, higher *δ*^13^C values were recorded at sites located in lakes (Figure 1, Mann–Whitney U test, *p* < 0.05), whereas increased *δ*^15^N values were found in specimens growing in rivers (Figure 2, Mann–Whitney U test, *p* < 0.05).

### 2.2. Relationships Between Water Physicochemical Variables and Isotopic Signal of Plants

The correlation analysis (Figure 3) revealed distinct patterns for *δ*^13^C and *δ*^15^N in relation to environmental variables. The *δ*^13^C values were positively correlated with dissolved oxygen (*p* < 0.05), pH (*p* > 0.05), depth (*p* < 0.001), and sediment organic matter and C/N ratio of plants (*p* < 0.001) and negatively correlated with total nitrogen and total phosphorus, bicarbonate, conductivity, DOC, and Ca^2+^ concentration (all *p* < 0.01–0.001), indicating that *δ*^13^C reflects both oxygenation and carbon source dynamics. In contrast, *δ*^15^N showed strong positive correlations with total nitrogen, bicarbonate, conductivity, and DOC and negative correlations with O_2_ concentration, depth, and sediment organic matter (*p* < 0.05–0.001), underscoring its sensitivity to nitrogen enrichment and organic matter load.

Based on Principal Component Analysis (PCA) supported by Spearman correlation analyses (Figure 3 and Figure 4), a negative relationship between *δ*^13^C and *δ*^15^N can be observed. In the case of nitrogen, a noticeable relationship was found between *δ*^15^N and phosphorus concentration, which indicated a more eutrophic condition. Additionally, *δ*^15^N showed positive correlations with calcium ions, bicarbonates, and dissolved organic carbon (Figure 3 and Figure 4), suggesting that increases in these parameters may reflect additional substances in the water and its mineralization. These relationships were aligned with the first PCA axis, which explained 32% of the total variance. In contrast, *δ*^13^C was positively associated with water depth and dissolved oxygen concentration along the same PCA axis. The second PCA axis explained less variance (10.2%) and was primarily associated with positive correlations with CO_2_ and CO_3_^2−^ concentrations (Figure 4). Notably, the water parameters and isotopic signatures showed substantial differentiation between the two studied aquatic ecosystem types (Figure 4).

## 3. Discussion

### 3.1. δ^13^C and δ^15^N Values of Submerged Macrophytes from Rivers and Lakes as Indicators of Environmental Change in Aquatic Ecosystems

Higher macrophyte *δ*^15^N values are correlated with increased total phosphorus (TP), suggesting that they may indicate the input of substances contributing to the eutrophication of inland waters. Such pollution can originate from fertilizers or result from runoff carrying contaminants from urban and rural areas [13]. Identifying pollution sources through the analysis of *δ*^15^N values is essential for initiating effective conservation measures.

Currently, programs are being implemented in Poland to enrich agricultural soils with calcium in order to reduce soil acidity [44]. Agricultural practices can result in soil erosion and the movement of salts, including calcium, into groundwater through surface runoff. Given the observed positive correlation between *δ*^15^N values and calcium ion concentrations, *δ*^15^N may serve as a useful indicator for tracking environmental changes in aquatic systems associated with fertilization. Higher mean and median calcium concentrations were observed in rivers, which may be explained by their greater sensitivity to changes occurring within the catchment area. Unlike lakes, nutrient availability in rivers is strongly influenced by flow dynamics [29]. Their specific morphology enables them to function as natural channels for transporting pollutants from the surrounding catchment area [1,45]. For this reason, riverine environments require particular attention and continuous monitoring in order to minimize the impact of anthropogenic factors on their balance of ecosystems [4]. In the study by Pronin et al. [13], nitrogen isotope signatures in rivers were higher in catchments dominated by agricultural use and dense development compared to those in forested catchments.

In the conducted study, river sampling sites exhibited a higher mean vegetation cover than those in lakes, a finding that might initially seem inconsistent with the previously stated hypothesis of greater anthropogenic pressure on lotic systems compared to lentic ones. However, this pattern may reflect the enhanced capacity of rivers to remove pollutants through advective transport and hydrodynamic flushing. Conversely, lakes, due to their lower water exchange rates, tend to retain and accumulate contaminants in bottom sediments, a process that can also be interpreted as a mechanism of internal self-purification within aquatic systems [46]. Sediments rich in organic matter remain a reservoir of contaminants that can be released back into the environment under the influence of physicochemical processes, including changes in temperature and pH, as well as through sediment resuspension caused by water mixing and oxygenation [47,48].

Higher mean organic matter content in sediments was recorded at lake sites (Appendix A). Due to slower carbon cycling in lakes compared to rivers, aquatic plants may have a greater likelihood of assimilating heavier carbon isotopes present in the water. Rivers, through processes such as self-purification, organic matter transformation, and their linear hydrological structure, often maintain a larger pool of available carbon supplied by pollutant inputs. This can result in lower *δ*^13^C values in macrophytes growing in river systems.

The mean C/N ratio was lower in rivers, while the maximum value was three times higher compared to lakes. This ratio showed a positive correlation with *δ*^13^C. Such a relationship may reflect differences in the availability and uptake of inorganic carbon by macrophytes across different types of aquatic ecosystems. The study by Pronin et al. [36] indicates that the C/N ratio in plant tissues serves as a significant indicator of the trophic status of the environment. Thus, the lower C/N ratio values in rivers may suggest higher nitrogen availability and more intensive assimilation of this element. The positive correlation between the C/N ratio and *δ*^13^C supports the hypothesis that both parameters respond to similar environmental drivers. As demonstrated by Pronin et al. [12], photosynthetic processes play a significant role in carbon isotope fractionation, and when combined with analysis of the C/N ratio, they allow for a more precise identification of organic matter sources in bottom sediments. In riverine ecosystems, characterized by higher flow variability and dynamic nutrient transport, patterns of carbon and nitrogen accumulation in plant tissues differ from those observed in lakes. This study’s findings suggest that the combined use of C/N ratios and stable isotope signatures is an effective tool for monitoring the movement of organic matter between terrestrial and aquatic systems, particularly in the context of anthropogenic pressure and land use changes within the catchment.

### 3.2. Influence of Physicochemical Indicators on the Formation of Isotopic Signatures in Macrophytes

Eutrophication or intense photosynthetic activity under conditions of low CO_2_ concentration may lead to reduced isotopic fractionation, resulting in higher carbon isotope signature values compared to environments with greater CO_2_ availability [42]. As the lighter carbon isotope becomes depleted, the likelihood of heavier isotope assimilation increases, while the capacity for preferential uptake of ^12^C diminishes, leading to an increase in *δ*^13^C values [12]. This phenomenon is more complex. Allochthonous organic matter, supplied from the catchment or introduced through localized pollution, contributes lighter carbon isotopes to lakes and rivers. Through mineralization, this lighter carbon becomes available for uptake by aquatic plants [49].

The *δ*^13^C values are influenced by pH, which is linked to shifts in the dominant form of dissolved inorganic carbon (DIC) under different acidity conditions. The *δ*^13^C is largely shaped by the preferential uptake of the lighter carbon isotope by aquatic plants. Therefore, elevated *δ*^13^C values in lakes compared to rivers may indicate a depletion of the lighter isotope or more intensive assimilation of the isotopically heavier DIC form, such as HCO_3_^−^ [14]. Correlations associated with site depth indicate changes in isotope ratios along the depth gradient. This gradient is characterized by temperature variation, which directly influences the solubility of certain gases and, consequently, their availability for uptake by aquatic plants [1,46].

External factors, such as the characteristics of groundwater, may have a greater influence on the formation of macrophyte isotope signatures than internal factors like growth rate. Similar patterns are reported in the study by [40], where the characteristics of the waterbody had a stronger influence on *δ*^13^C and *δ*^15^N values than the percentage content of carbon, nitrogen, and phosphorus in the leaves of the analyzed macrophytes. In the study by Chappuis et al. [40], emphasis was placed on the role of the functional group to which a plant belongs (e.g., submerged macrophytes, floating-leaved species, isoetids, and charophytes). This observation corresponds with the findings presented by Pronin et al. [12,36], where a significant influence of both functional group affiliation and photosynthetic mechanism on isotopic signatures was demonstrated. It determines the plant’s ability to acquire carbon from the atmosphere but also defines the surface area available for carbon uptake from either water or air [50]. Similar to the findings presented in this study for *P. perfoliatus*, *S. pectinata*, and *P. crispus*, it was observed that fully submerged vegetation exhibits more pronounced differentiation in its carbon isotope signature. This enhances its capacity to reflect environmental contrasts and capture ongoing changes within aquatic systems.

### 3.3. Physiological Factors Influencing the Variability of the δ^13^C Isotope Signal in Submerged Macrophytes

The formation of *δ*^13^C is influenced not only by the carbon source but also by the mechanism of active bicarbonate uptake. In this process, carbonic anhydrase (CA) plays an important role. This enzyme facilitates the rapid conversion of bicarbonate ions into carbon dioxide, which is then directly utilized in sugar production through photosynthesis [21]. Bicarbonate ions exhibit higher *δ*^13^C values than dissolved carbon dioxide, which may lead to elevated *δ*^13^C in plants that rely on this carbon source under conditions of limited CO_2_ availability. On the other hand, although carbonic anhydrase does not itself fractionate isotopes, it facilitates the rapid interconversion between HCO_3_^−^ and CO_2_. This enzymatic activity enables isotopes ^12^C and ^13^C to fractionate according to their natural physicochemical properties, thereby influencing the resulting *δ*^13^C values in plant tissues [51]. As a result, this enzyme contributes to achieving isotopic equilibrium, and differences in isotope ratios can be used to infer environmental changes [51]. This enzyme is found, among others, in the genus *Potamogeton*.

It is important to note that the Principal Component Analysis (Figure 3) captures only a limited portion of the relationships among the variables examined in this study. Approximately 60% of the variability remains unexplained. Certain phototrophic microorganisms, equipped with their own carbonic anhydrase, influence the composition of the available carbon isotope pool. At the same time, biofilm interactions involving the production of organic acids alter pH at the microscale within the local environment, affecting the forms of carbon present. Microbial decomposition of organic matter increases the proportion of ^12^C in the surrounding water [52]. The role of microorganisms in shaping the microscale habitat of macrophytes is not yet fully understood, including the strength of this interaction, yet evidence suggests that such a relationship exists.

In addition, certain groups of aquatic plants are capable of absorbing CO_2_ from the pore spaces within sediments, a process facilitated by their well-developed root systems [53]. In softwater lakes, which are characterized by low CO_2_ concentrations but still the most dominant form of C, there are groups of macrophytes that exhibit alternative photosynthetic adaptations. Some isoetids store carbon at night in the form of organic acids and perform CAM (Crassulacean Acid Metabolism) photosynthesis. This strategy reduces competition for carbon during the day, when other species are actively photosynthesizing and thus assimilating carbon most intensively [50]. This carbon uptake strategy will significantly affect *δ*^13^C, as shown by studies by Pronin et al. [36]. In the referenced study, the authors documented that plants employing different photosynthetic metabolic pathways (C_3_, CAM, and those with a carbon concentrating mechanism (CCM)) exhibit distinct ranges of carbon isotope signatures. CAM plants show intermediate *δ*^13^C values compared to C_3_ and CCM species, which directly reflects their adaptation to carbon uptake under conditions of limited CO_2_ availability. Pronin et al. [36] also observed that the variability of isotopic signatures is strongly influenced by habitat conditions, particularly the DIC concentration and composition, which explains the observed differences between riverine and lacustrine ecosystems.

The present study focused on species utilizing the C_3_ photosynthetic pathway, which relies on the enzyme Rubisco involved in the Calvin cycle for CO_2_ fixation. Rubisco preferentially assimilates the lighter isotope ^12^C over ^13^C, resulting in strong carbon isotope fractionation. Consequently, *δ*^13^C values typically range from −24‰ to −30‰ relative to the PDB standard (currently V-PDB). C_4_ plants, which use PEP carboxylase (an enzyme with reduced selectivity for ^12^C), exhibit less pronounced isotopic fractionation. As a result, they show higher *δ*^13^C values, typically ranging from −10‰ to −14‰. Similar values are observed in macrophyte groups possessing CCM that depend primarily on HCO_3_^−^ as a carbon source. Macrophytes employing CAM photosynthesis tend to exhibit higher *δ*^13^C values on average compared to C_3_ plants, depending on the intensity of CAM expression. However, these values show less variation than those observed in terrestrial vegetation, where *δ*^13^C typically ranges from −10‰ to −20‰ [54].

The *δ*^13^C values obtained for the macrophytes examined in this study fell within the range typical of C_3_ plants [55]. However, significant interspecific differences were observed, which can be attributed to variation in carbonic anhydrase activity and the capacity to assimilate carbon from different sources. Particularly notable differences were found among species of the genus *Potamogeton*, supporting earlier reports on the important role of carbonic anhydrase in carbon acquisition in these plants [56]. These findings indicate that the carbon isotopic signal in the tissues of submerged macrophytes results from complex interactions among physiological carbon uptake mechanisms, the availability of various forms of inorganic carbon in the environment, and microbial activity in the root zone, as described in previous studies [13,36].

### 3.4. An Ecophysiological and Ecosystem-Level Approach to δ^15^N

A thorough examination of factors such as nitrogen sources, uptake mechanisms, and environmental conditions is essential to the interpretation of *δ*^15^N in ecological studies. Analysis of these processes can begin at the atomic scale. As observed with carbon, bonds formed by lighter nitrogen isotopes break or form more quickly than those involving heavier isotopes. During enzymatic reactions, this difference causes lighter isotopes to be processed more easily, leaving the substrate enriched in the heavier isotope and the product enriched in the lighter one. This isotopic discrimination produces characteristic *δ*^15^N values [57]. An example of this is the reduction of nitrate by nitrate reductase, which is associated with a significant isotope effect of approximately 16‰. Other processes, such as nitrate absorption and transport across cell membranes, also exhibit minor isotope effects, around 1.003‰. The final isotopic signal is influenced by several factors, including the ambient nitrogen concentration, its chemical form, and the specific type of nitrate reductase involved [57]. The phenomenon of differing isotope effects may apply not only to the plant itself but also to the surrounding microbiome that utilizes the same enzyme, thereby altering the pool of available isotopes through selective uptake or transformation.

The specific plant organ subjected to isotopic analysis also plays a significant role. The *δ*^15^N analysis in different parts of the plant provides additional information about internal nitrogen transport and metabolism, which affects the choice of the tissue to be examined [58,59]. Glutamine synthetase, glycine decarboxylase, and transaminases are involved in later stages of nitrogen metabolism, including ammonia assimilation and amino acid synthesis. The reactions catalyzed by these enzymes can also lead to isotopic fractionation, influencing *δ*^15^N values of specific metabolites in different plant organs [58,59]. The internal establishment of isotopic values represents an important adaptive mechanism, enabling plants to optimize the use of available nitrogen resources [57].

Although plant physiology plays a major role in shaping isotopic values, most of the relationships observed in our study are linked to environmental factors, particularly the immediate surroundings of the plant. Using isotopic techniques, it has been demonstrated that *δ*^15^N values can serve as indicators of eutrophication and sources of nitrogen pollution [20,49,60]. The *δ*^15^N values in macrophytes growing near municipal wastewater inflows were higher than those in less polluted areas. Pronin et al. [20], in their comprehensive isotopic analysis of macrophytes from lakes in northern Poland, also observed a strong correlation between *δ*^15^N values and the degree of anthropogenic pressure. The highest *δ*^15^N values were recorded in plants absorbing nitrogen from sediments impacted by runoff from urbanized areas, where the dominant nitrogen forms were organic compounds undergoing intensive mineralization. Similar research was conducted on Lake Titicaca, which is subject to intense anthropogenic pressure, including significant pollutant inflow. The results, consistent with the findings presented in this study, show an increase in *δ*^13^C values and a decrease in *δ*^15^N values in macrophytes and suspended organic matter with increasing distance from the pollution source [49]. The interdependence between the biogeochemical cycles of carbon and nitrogen is fundamental to understanding macrophyte adaptation to changing environmental conditions and their role in the functioning of aquatic ecosystems. Under conditions of elevated nitrogen concentrations, macrophytes adopt various strategies, often leading to a significant reduction in fresh biomass production and a decrease in total chlorophyll content [60]. High nitrogen concentrations can also lead to growth inhibition as a result of the accumulation of free amino acids produced during detoxification processes [60]. The use of ammonium as a nitrogen source is associated with higher *δ*^15^N values. At the same time, lower *δ*^13^C values are often observed under similar conditions, suggesting a shift in the source of carbon uptake as well [49].

The results obtained, in light of the studies discussed above, highlight the role of macrophytes as specific “recorders” of biogeochemical processes occurring within ecosystems. This implies that the isotopic composition of macrophyte tissues reflects not only the momentary concentrations of various nitrogen forms but also long-term trends and processes, integrating information on nitrogen sources, transformations, and cycling within the aquatic environment [12,13].

## 4. Materials and Methods

### 4.1. Study Area

This study used plant material collected from 2008 to 2011 by researchers from the Department of Plant Ecology, Faculty of Biology, University of Gdańsk. The herbarium material provided was collected from 38 sites located on 15 rivers (16 localizations, including 2 sites located on different sections of the same river) and 108 sites within 21 lakes in northern Poland (Figure 5). The studied pondweed species, i.e., *P. perfoliatus*, *P. crispus*, and *S. pectinata*, were collected from the following lakes: 1. Choczewskie, 2. Raduńskie Dolne, 3. Wdzydze, 4. Długie, 5. Ińsko, 6. Łąckie, 7. Raduńskie Górne, 8. Sominko, 9. Trzcińskie, 10. Dymno, 11. Piaszno, 12. Żarnowieckie, 13. Kotel, 14. Borzyszkowskie, 15. Płocice Wielkie, 16. Garczyn, 17. Kamieniczno, 18. Sczuczarz, 19. Trzebielsk, 20. Gardliczno, 21. Prusionki Małe.

In addition, pondweeds were collected from the following rivers: 1. Brda in Męcikał town, 2. Chocina, 3. Cieplicówka, 4. Kępiny Małe (ditch), 5. Linawa, 6. Nogat in Jazowa village, 7. Studnica, 8. Szkarpawa, 9. Tuga in Żelichowo village, 10. Wisła, 11. Wda in Bałachy village, 12. Zagórska Struga, 13. Zbrzyca in Laska village, 14. Pilica in Łubiana village, 15. Wisła Królewiecka, and 16. Wda in Borsak village (Figure 5). The sites selected for this study were chosen based on the availability of archived material from a larger study on pondweed species ecology, previously published by [28].

### 4.2. Physicochemical Analysis of Water and Sediment Parameters

During extensive field study, various parameters were measured, including water pH, conductivity, O_2_, photosynthetically active radiation (PAR), species coverage, and depth. Water conductivity was measured with an LF 96 conductivity meter (WTW, Germany) with a Tetracon 96 electrode; pH and redox potential with a WTW 320/SET 1 pH meter (WTW, Weilheim, Germany) using Mettler (Greifensee, Switzerland) and Sentix 97T (WTW, Weilheim, Germany) electrodes. Dissolved inorganic carbon (CO_2_, HCO_3_^−^, and CO_3_^2−^) was quantified by titration, and calcium by EDTA complexometry. PAR was measured using a Licor LI-250 Light Meter (LI-COR Environmental GmbH, Bad Homburg, Germany) and is expressed here as a percentage. The species coverage was investigated by diving, and its percentage cover in the sites (0.1 m^−2^ of the bottom) included in the analyses was estimated. Total nitrogen (TN) and total phosphorus (TP) concentrations were determined using photometric methods with the MERCK Spectroquant cuvette test on the UV–VIS spectrophotometer (Aquamate, Thermo Electron Corporation, Waltham, MA, USA). Therefore, TP analysis was performed after mineralizing water samples in a mixture of sulfuric and nitric acids in a 2:1 proportion in the microwave digestion system Mars 5 CEM (Matthews, NC, USA). The concentration of humic acids was analyzed by treating them as dissolved organic carbon (DOC), following the methodology described in Merdalski et al. [28] using a UV–vis spectrophotometer (Aquamate, Thermo Electron Corporation, Waltham, MA, USA) at a wavelength of 330 nm. Additionally, in sediment samples collected in the field (50 cm^3^), the amount of organic matter (OM) was determined using the loss-on-ignition method at 550 °C (Thermolyne 62700 muffle furnace, Waltham, MA, USA). The aforementioned analyses enable the investigation of *δ*^13^C and *δ*^15^N in macrophytes in the context of their ecological niche.

### 4.3. Isotopic Analysis of Plant Material

The collected plant samples were washed in the field, and the epiphytes and other contaminations (sand, sediments, and others) were removed and washed using deionized water. The plant samples were dried at 60 °C for 48 h and stored dry. The plant materials stored in a dry state were archived in paper envelopes until 2022. The isotopic analysis was then conducted at the GISMO Isotope Laboratory at the University of Burgundy in Dijon, France. To prepare the samples for analysis, the plant material was first ground using a ball mill (MM 400 Retsch, Haan, Germany). For small sample quantities, an agate mortar was used for grinding. The next step involved the removal of carbonates. This was achieved using the desiccator method, leveraging the reaction of carbonates with 37% HCl, which produced an acidic mist that evaporated from the test tubes in a fume hood. After 48 h in the desiccator, the plant material was placed in small glass vials under the fume hood for an additional 24 h to ensure complete evaporation. Subsequently, the material was dried in an oven at 40 °C for 24 h. To obtain the most homogeneous fine powder, the samples were reground using an agate mortar. Following this, the processed material was placed into tiny capsules, weighed, sealed, and analyzed using a Flash Smart EA elemental analyzer (Thermo Scientific, MA, USA) connected to a Delta V stable isotope ratio mass spectrometer (Thermo Scientific, MA, USA). For calibration and quality control, the USG40 standard (glutamic acid, *δ*^13^C = −26.39‰, *δ*^15^N = −4.5‰) and standard wheat flour B2157 (Elemental Microanalysis, Okehampton, UK) were used. The *δ*^13^C and *δ*^15^N values were expressed in per mil (‰) relative to the V-PDB standard for carbon and atmospheric N_2_ for nitrogen. The precision of the analysis was confirmed through external reproducibility of repeated standard measurements (USG40 and B2157), demonstrating accuracy better than ±0.15‰ for *δ*^13^C and ±0.20‰ for *δ*^15^N (2σ).

Additionally, the elemental analyzer Flash Smart EA (Thermo Scientific, Waltham, MA, USA) provided measurements of the percentage of carbon (% C in OM_PLANTS_) and nitrogen in the organic plant matter (% N in OM_PLANTS_), enabling the calculation of the carbon-to-nitrogen ratio (C/N).

The standard formula for calculating isotopic signatures, considering the isotope ratios of ^13^C/^12^C and ^15^N/^14^N, is presented in Equation (1).(1)$$\delta \left(‰\right)=\frac{R_{sample}{-R}_{standard}}{R_{standard}}\times 1000$$
where R_sample_ is ^13^C/^12^C for carbon and ^15^N/^14^N for nitrogen in analyzed samples, while R_standard_ is ^13^C/^12^C for carbon and ^15^N/^14^N for nitrogen in analyzed standards.

### 4.4. Statistical Analysis

The analyses presented in this study were conducted using R software, version R.4.2.2 [61]. The packages used included *corrplot* [62], *ggplot2* [63], *FactoMineR* [64], *factoextra* [65], and *dunn.test* [66]. Initially, the isotopic signature values were tested for normality using the Shapiro–Wilk test. Since the data did not meet the assumption of a normal distribution, non-parametric analyses were applied. The Mann–Whitney U test (U-M H) was applied to assess significant differences between the two analyzed aquatic environments, i.e., lakes and rivers. Spearman’s rank correlation analysis, visualized below as a correlation heatmap, allowed for the evaluation of relationships between *δ*^13^C, *δ*^15^N, and environmental variables included in this study. The Kruskal–Wallis H test (K-W H) was used to determine whether *δ*^13^C and *δ*^15^N values differed among submerged plant species. Post hoc Dunn’s test was applied to identify specific differences in isotopic signatures between species when statistically significant differences were detected using the K-W H test. Additionally, Principal Component Analysis (PCA) was used to graphically represent relationships between environmental variables and the *δ*^13^C and *δ*^15^N values of the studied species. PCA is a widely known and fundamental multivariate statistical technique for dimensionality reduction and for visualizing patterns in multivariate data [67], which we used to check which, included in this study, environmental parameters are strictly related to *δ*^13^C and *δ*^15^N values of investigated macrophytes.

## Figures and Tables

**Figure 1 plants-14-02261-f001:**
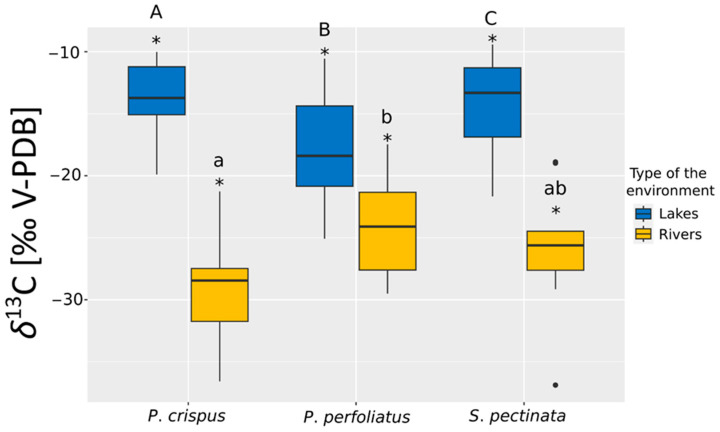
Comparison of the *δ*^13^C value distribution among the three studied submerged plant species: *P. crispus*, *P. perfoliatus*, and *S. pectinata* in lakes and rivers. Letters A, B, and C are for lakes, and letters a and b are for rivers; statistically significant differences between the studied species according to the Kruskal–Wallis (K-W H; *p* < 0.05) and Dunn’s tests (*p* < 0.05) are for the identified specific differences. An asterisk (*) means statistically significant differences between values for lakes and rivers according to the Mann–Whitney U test (U M-W; *p* < 0.05). Boxes represent 25th–75th percentiles, with horizontal lines indicating medians, whiskers indicating the minimum and maximum values (excluding outliers), and black dots representing extreme results.

**Figure 2 plants-14-02261-f002:**
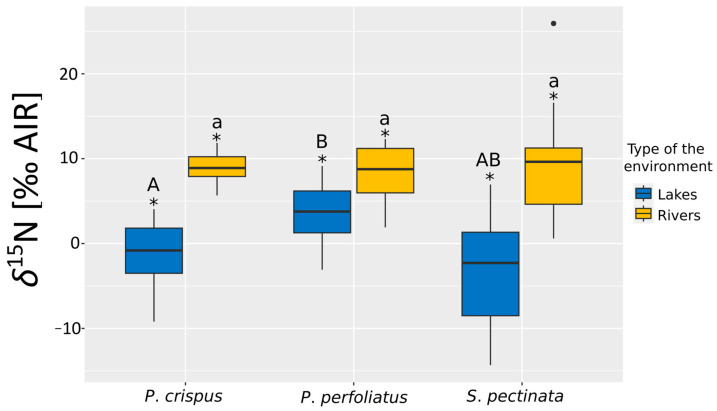
Comparison of the *δ*^15^N value distribution among the three studied submerged plant species: *P. crispus*, *P. perfoliatus*, and *S. pectinata* in lakes and rivers. Letters A and B are for lakes, and they indicate statistically significant differences between the studied species according to the Kruskal–Wallis (K-W H; *p* < 0.05) and Dunn’s tests (*p* < 0.05); letter a stands for rivers, identifying non-statistically significant differences. An asterisk (*) means statistically significant differences between values for lakes and rivers according to the Mann–Whitney U test (U M-W; *p* < 0.05). Boxes represent 25th–75th percentiles, with horizontal lines indicating medians, whiskers indicating the minimum and maximum values (excluding outliers), and black dots representing extreme results.

**Figure 3 plants-14-02261-f003:**
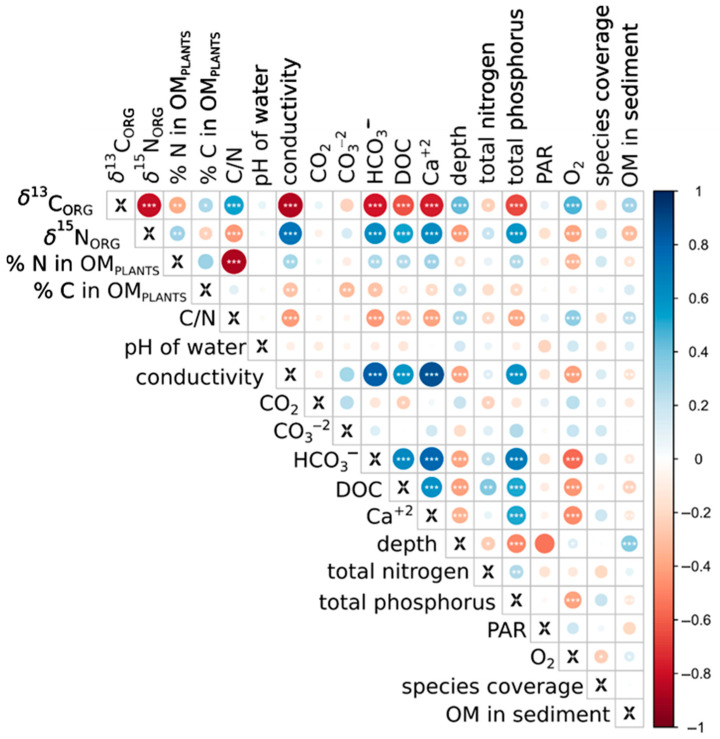
Spearman correlation heatmap for *δ*^13^C and *δ*^15^N values of organic matter across all studied sites of the three macrophyte species (*n* = 146) and the analyzed environmental variables. OM—organic matter; DOC—dissolved organic carbon; PAR—photosynthetically active radiation. *, **, and *** indicate statistical significance on the <0.05, <0.01, and <0.001 levels, respectively.

**Figure 4 plants-14-02261-f004:**
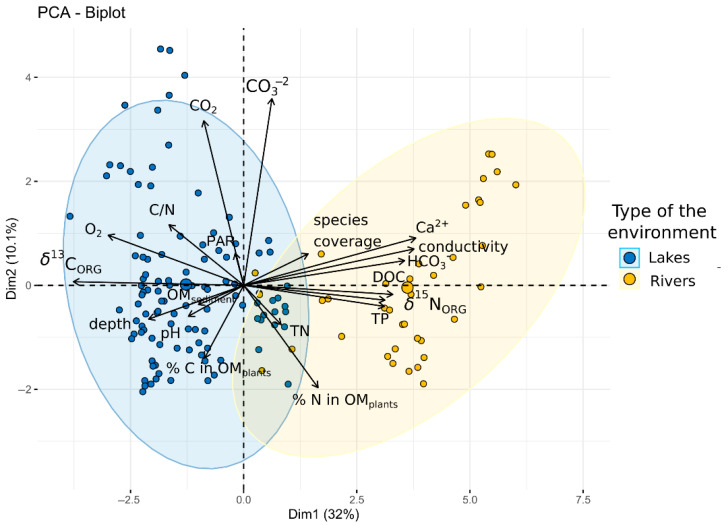
Principal Component Analysis (PCA) for *δ*^13^C and *δ*^15^N values across all studied sites of the three macrophyte species from lakes and rivers, along with the analyzed environmental variables. DOC—dissolved organic carbon; PAR—photosynthetically active radiation; TP—total phosphorus; TN—total nitrogen.

**Figure 5 plants-14-02261-f005:**
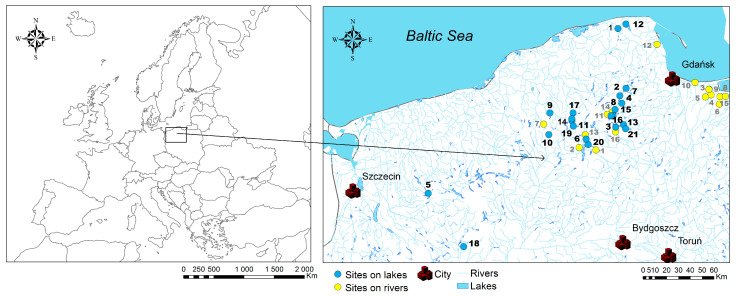
Location of study sites. The number corresponded to the order listed in the text (black numbers correspond to lakes and gray numbers correspond to rivers).

## Data Availability

Most of the data generated or analyzed during this study are included in this article and its Appendix A. The authors will make the rest of the included data available upon reasonable request.

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
