# Peer review of "Stable Carbon and Nitrogen Isotope Signatures in Three Pondweed Species—A Case Study of Rivers and Lakes in Northern Poland"

_plants, 2025, doi:10.3390/plants14152261_

Round 1
Reviewer 1 Report
Comments and Suggestions for Authors
This article is worth publishing.
However, some questions should be addressed.
The explanation of why these three species of Potamogetonaceae should be added.
P. 3, lines 108-109, “It is important to highlight that rivers and lakes differ significantly in their susceptibility to pollution and in the intensity of environmental pressure they experience” – Strong difference in water retention time between lentic and lotic ecosystems and continuous formation of new waters physically replacing and partially mixing with older ones in flowing water appear to be more reasonable explanation in comparison to the explanation described in Introduction.
P. 3, lines 110-111, “In flowing waters, once pollutant inputs stop, the ecosystem has the capacity to recover through natural self-purification processes” – A more easy explanation is the physical replacement of polluted water with new parcels of clearer water.
Could you please add an explanation of the difference in nutrient availability between rivers and lakes?
P. 3, lines 121-122 – Could you please explain why “water flow characteristics” are listed together with “hydrological conditions”? It appears that hydrological conditions, by definition, include water flow characteristics.
Figure 1. – “c” is missing in the figure in contrast to its legend. Could you please add the meaning of black dots in the right part of the figure?
Figure 2. – “C”, “b”, “c” are missing in the figure in contrast to its legend. Could you please add the meaning of the black dot in the right part of the figure?
P. 11, line 409, “(15 localization)” – Could you please explain the meaning of localization in this context?
Figure 3. Some cells appear empty because small non-contrast dots in them are not far from invisible. Could you please improve this figure?
The coordinates of sampling sites and dates of collection should be added as a table in the supplement.
Figure 4. The figure showing borders between states in Europe should be updated. In the current version, it denies the existence of Montenegro as a worldwide internationally recognized state.
Could you please add an explanation of how these sites were chosen for the study? According to Figure 4, many river sites are situated east of Gdańsk in contrast to lake sites, mostly situated far west of Gdańsk. It might affect the results of this study because of differences in natural background and land use. Could you please add an explanation of why the geographical position of these sites was not crucial for the aim of this study?
P. 12, lines 426-427 - A brief explanation, including the equipment and techniques used for measuring water pH, conductivity, O2, photosynthetically active radiation (PAR), and species coverage, should be added.
P. 12, line 437 – A brief explanation about the way of drying the collected plant material should be added.
Author Response
Comment 1. This article is worth publishing.
Response 1. Thank you very much for enthusiastic comments to our study.
Comment 2. However, some questions should be addressed.
The explanation of why these three species of Potamogetonaceae should be added.
Response 2. Thank you for this suggestion. The sentence explains why these three species were added.
The selected species from the Potamogetonaceae family were chosen due to their differing habitat requirements, their representative presence in Potamogeton-type lakes, and their frequent occurrence in lowland rivers of Poland.
Comment 3. P. 3, lines 108-109, “It is important to highlight that rivers and lakes differ significantly in their susceptibility to pollution and in the intensity of environmental pressure they experience” – Strong difference in water retention time between lentic and lotic ecosystems and continuous formation of new waters physically replacing and partially mixing with older ones in flowing water appear to be more reasonable explanation in comparison to the explanation described in Introduction.
Response 3. Thank you for this suggestion. We modified the sentences to highlight also this simple explanation which Reviewer 1 suggested.
Comment 4. P. 3, lines 110-111, “In flowing waters, once pollutant inputs stop, the ecosystem has the capacity to recover through natural self-purification processes” – A more easy explanation is the physical replacement of polluted water with new parcels of clearer water.
Response 4. Thank you for this suggestion. We modified the sentences to highlight also this simple explanation which Reviewer 1 suggested.
Comment 5. Could you please add an explanation of the difference in nutrient availability between rivers and lakes?
Response 5. Thank you for this suggestion. We added the sentences to highlight also those dependencies.
Comment 6. P. 3, lines 121-122 – Could you please explain why “water flow characteristics” are listed together with “hydrological conditions”? It appears that hydrological conditions, by definition, include water flow characteristics.
Response 6. Thank you for the suggestion. We agree that this sentence's construction might be confusing, but we understand the hydrological conditions as an umbrella term referring to the state and dynamics of water in an ecosystem. It includes precipitation, groundwater inputs, surface runoff, water level, seasonal fluctuations, residence time, and flow regime. However, water flow characteristics are a subset of hydrological conditions, specifically describing the velocity, direction, turbulence, discharge rate, and variability of flowing water. Thus, we think that we left these sentences as it is.
Comment 7. Figure 1. – “c” is missing in the figure in contrast to its legend. Could you please add the meaning of black dots in the right part of the figure?
Response 7. Thank you for the suggestion. We used a standard description and did not notice that only two letters were used to indicate the observed differences. Therefore, the letter “c” has been removed from the figure caption.
Comment 8. Figure 2. – “C”, “b”, “c” are missing in the figure in contrast to its legend. Could you please add the meaning of the black dot in the right part of the figure?
Response 8. Thank you for the suggestion. We used a standard description and did not notice that only two letters were used to indicate the observed differences. Therefore, the letters “C,” “b,” and “c” have been removed from the figure caption. Additionally, the description was revised to explain the construction of the box-and-whisker plot and to indicate that the black dots represent extreme values.
Comment 9. P. 11, line 409, “(15 localization)” – Could you please explain the meaning of localization in this context?
Response 9. While adding the geographical coordinates, we noticed that one localization had not been marked on the map and was also omitted from the list of localizations in the manuscript text. We have now corrected this oversight. As a result, there is one additional localization in the list of river sites. Consequently, two sampling points are situated on the same river, but at different sites, approximately 30–40 km apart.
Comment 10. Figure 3. Some cells appear empty because small non-contrast dots in them are not far from invisible. Could you please improve this figure?
Response 10. As this figure body was automatically generated by an R package, we would prefer to retain its original design.
Comment 11. The coordinates of sampling sites and dates of collection should be added as a table in the supplement.
Response 11. Thank you for this suggestion the coordinates and the dates of the samples collection were provided.
Comment 12. Figure 4. The figure showing borders between states in Europe should be updated. In the current version, it denies the existence of Montenegro as a worldwide internationally recognized state.
Response 12. Thank you for your suggestion. For the map preparation, now we used the most recent shapefile available in ArcGIS. We sincerely apologize for the earlier old version of the Europe map
Comment 13. Could you please add an explanation of how these sites were chosen for the study?
Response 13. The sites selected for this study were chosen based on the availability of archived material from a larger study on Potamogetonaceae ecology, previously published by the co-authors of this work: Merdalski et al. 2019: Environmental Factors Affecting Pondweeds in Water Bodies of Northwest Poland. Biodivers. Res. Conserv. 2019, 56, 13–28, doi:10.2478/biorc-2019-0014.
Comment 14. According to Figure 4, many river sites are situated east of Gdańsk in contrast to lake sites, mostly situated far west of Gdańsk. It might affect the results of this study because of differences in natural background and land use. Could you please add an explanation of why the geographical position of these sites was not crucial for the aim of this study?
Response 14. Thank you for the suggestion. To be precise, approximately half of the investigated river sites are located near the lakes included in this study. We agree that land use differences among the sites may influence water chemistry, which in turn could affect the final isotopic signature of macrophyte organic matter. In our study, such an effect was observed only for the river sites, particularly those located in more urbanized areas (Pronin et al. 2025: Following the Footsteps of Macrophytes: Potential Application of Isotope Signals in Pollution Monitoring: A Case Study of Northern Polish Rivers. Ecohydrol. Hydrobiol. 2025, 100650, doi:10.1016/j.ecohyd.2025.100650. In contrast, the sites situated east of Gdańsk are more agricultural, as they lie on soils developed in the former delta of the Vistula River, which was diked and regulated centuries ago
Comment 15. P. 12, lines 426-427 - A brief explanation, including the equipment and techniques used for measuring water pH, conductivity, O2, photosynthetically active radiation (PAR), and species coverage, should be added.
Response 15. Thank you for the suggestion. The brief description was provided in this section.
Comment 16. P. 12, line 437 – A brief explanation about the way of drying the collected plant material should be added.
Response 16. Thank you for the suggestion. The brief description was provided in this section
Reviewer 2 Report
Comments and Suggestions for Authors
Comments
The manuscript entitled “Stable Carbon and Nitrogen Isotope Signatures in Three Pond-weed Species – A Case Study of Rivers and Lakes in Northern Poland” is an interesting work and has a significant contribution to the scientific community. Generally, the manuscript is in good condition, however several modifications are needed in order to be suitable for publication. Specifically:
- Abstract part: the abstract needs substantial modifications. Its introduction should be more general. I recommend changing the order of the sentences “ Since ….monitoring” (line 11-12) with lines 12-14 “ Aquatic ……pressure” . Also, you shouldn’t divide abstract into paragraphs. Additionally, you should be more specific about the anthropogenic influences of stable carbon (δ13C) and nitrogen (δ15N) isotope and give with which human activities are related.
- Line 34: “ of these environments” which environments? Please articulate
- Line 35: please remove the word “ other”
- Line 43: please be specific and add “ European Union's (EU)” before the water
- Line 46-48: reference is needed here
- Line 59-60: “ The greatest….alterations” some more information/examples about the water quality status of surface waters in Poland would be interesting & the some associated references
- Line 73: please use the associated abbreviation
- Line 74-75: “Eutrophication….phenomenon” please remove these lines, since are not contributing and doubtable
- Line 76-79: these lines are not blending well with the rest of the paragraph, please remove them.
- Line 59-73: you need to enrich your bibliographical sources for justification of this paragraph and not limited only to references 12-14.
- Line 110-111: “ In flowing… processes” reference is needed here
- Line 119: “ Based on a review of the literature and preliminary observations” please add some studies you used. Also, I noticed that you use no examples of other relevant/similar studies related to macrophytes in your manuscript. Please cite and briefly analyze some case studies from other researchers (preferably from Poland and Europe with similar climatological conditions) .
- Line 117-133: you must clearly and analytically explain the contribution of your study to limnological/water quality sciences and the community. Why is your work presented here useful and how it differs from similar studies? What’s the novelty of your work (if any)? Please highlight it.
- Line 134: why are you presenting results into part 2 instead of methodology? Please follow the journal’s guideline
- Line 136-169: the presentation of the results is not sufficient and needs to be enriched. For example, you use no specific information about the statistically significant differences in the main text, while the figure 1 legend contains some information about, which should be also included in the main text. Additionally, figure 1 is not described sufficiently in the legend, since e.g. , the horizontal line in the boxplots is not defined as what is presenting/symbolizing (the same for the vertical lines). Please apply the same comment for the rest of the figures using boxplots.
- Line 170-171: the results presentation is poor. Need to add more text and explain the negative correlation as well. At least for the strong and moderate correlations (positive and negative)
- Line 174-176: the figure 3 legend needs to be improved. Please say why some parameters have three dots in the circles or one? What the dots in the circle symbolize? Also, the parameters abbreviations should be included in the methodology part, which should be part 2.
- Line 177-181: also here must enrich the PCA results presentation
- Discussion part: generally good but need to add more references. Some whole paragraphs have no scientific justification.
- Line 408: department of what? University/ministry?
- Line 426-435: you need to explain what is corresponding to each abbreviation you use.
- Line 463: the equ 1 should be following the mathematical formula
- Line 464: the definitions of the equation symbols should be more carefully defined
- Line 479-481: “ Additionally, Principal Component Analysis (PCA) was used to graphically represent relationships between environmental variables and the δ13C and δ15N values of the 480 studied species.” This part needs to be modified. You should briefly explain what is PCA , based on scientific references and afterwards to say why is PCA useful for your data analysis. Also, some studies using PCA and Macrophyte/similar water quality parameters should be added in the introduction or discussion part and briefly analyze their findings.
Author Response
Comment 1. The manuscript entitled “Stable Carbon and Nitrogen Isotope Signatures in Three Pond-weed Species – A Case Study of Rivers and Lakes in Northern Poland” is an interesting work and has a significant contribution to the scientific community. Generally, the manuscript is in good condition, however several modifications are needed in order to be suitable for publication.
Response 1. Thank you very much for your enthusiastic comments on our study.
Comment 2. Specifically:
Abstract part: the abstract needs substantial modifications. Its introduction should be more general. I recommend changing the order of the sentences “ Since ….monitoring” (line 11-12) with lines 12-14 “ Aquatic ……pressure” . Also, you shouldn’t divide abstract into paragraphs. Additionally, you should be more specific about the anthropogenic influences of stable carbon (δ13C) and nitrogen (δ15N) isotope and give with which human activities are related.
Response 2. Thank you for this suggestion, the suggested Abstract modification were applied.
Comment 3. Line 34: “ of these environments” which environments? Please articulate
Response 3. Thank you for this suggestion. We mean here mentioned earlier lakes and rivers. The sentence was modified to be more precise.
Comment 4. Line 35: please remove the word “ other”
Response 4. Thank you for this suggestion. Removed.
Comment 5. Line 43: please be specific and add “ European Union's (EU)” before the water
Response 5. Thank you for this suggestion. Added.
Comment 6. Line 46-48: reference is needed here
Response 6. Thank you for this suggestion. Added
Comment 7. Line 59-60: “ The greatest….alterations” some more information/examples about the water quality status of surface waters in Poland would be interesting & the some associated references
Response 7. Thank you for this suggestion. Added
Comment 8. Line 73: please use the associated abbreviation
Response 8. Thank you for this suggestion. Done.
Comment 9. Line 74-75: “Eutrophication….phenomenon” please remove these lines, since are not contributing and doubtable
Response 9. Thank you for this suggestion. Done.
Comment 10. Line 76-79: these lines are not blending well with the rest of the paragraph, please remove them.
Response 10. Thank you for this suggestion. Done.
Comment 11. Line 59-73: you need to enrich your bibliographical sources for justification of this paragraph and not limited only to references 12-14.
Response 11. Thank you for this suggestion. Done.
Comment 12. Line 110-111: “ In flowing… processes” reference is needed here
Response 12. Thank you for this suggestion. Done.
Comment 13. Line 119: “ Based on a review of the literature and preliminary observations” please add some studies you used. Also, I noticed that you use no examples of other relevant/similar studies related to macrophytes in your manuscript. Please cite and briefly analyze some case studies from other researchers (preferably from Poland and Europe with similar climatological conditions).
Response 13. Thank you for this suggestion. Several references are added. However, we decided to shortly describe only this one most relevant to our study.
Comment 14. Line 117-133: you must clearly and analytically explain the contribution of your study to limnological/water quality sciences and the community. Why is your work presented here useful and how it differs from similar studies? What’s the novelty of your work (if any)? Please highlight it.
Response 14. Thank you for the suggestion. The sentences have been revised to emphasize the novelty and importance of this study. We added:
To best of our knowledge, this study is the first in Poland, and among the first in Europe, to compare δ13C and especially δ15N signatures of S. pectinata, P. perfoliatus, and P. crispus between river and lake environments. The results highlight clear habitat-related isotopic differences and demonstrate the potential of these species as sensitive bioindicators of eutrophication and nutrient pollution. These findings advance our understanding of the factors shaping isotopic variation in macrophytes and support their use in developing more effective approaches to monitoring aquatic ecosystem health.
Comment 15. Line 134: why are you presenting results into part 2 instead of methodology? Please follow the journal’s guideline
Response 15. This scheme was used in the journal template provided to authors, as well as in most previously published papers, including the recent article by Pronin et al. (2024), and it was approved by the editors
Comment 16. Line 136-169: the presentation of the results is not sufficient and needs to be enriched. For example, you use no specific information about the statistically significant differences in the main text, while the figure 1 legend contains some information about, which should be also included in the main text. Additionally, figure 1 is not described sufficiently in the legend, since e.g. , the horizontal line in the boxplots is not defined as what is presenting/symbolizing (the same for the vertical lines). Please apply the same comment for the rest of the figures using boxplots.
Response 16. Thank you for this suggestion. We added the legend explanation and the description of the differences in the main text.
Comment 17. Line 170-171: the results presentation is poor. Need to add more text and explain the negative correlation as well. At least for the strong and moderate correlations (positive and negative)
Response 17. Thank you for this suggestion. We added the condensed description of the correlations. We apologize for the earlier, too-short description.
Comment 18. Line 174-176: the figure 3 legend needs to be improved. Please say why some parameters have three dots in the circles or one? What the dots in the circle symbolize? Also, the parameters abbreviations should be included in the methodology part, which should be part 2.
Response 18. Thank you for the suggestion, they were applied, we explain the *, **, *** in the Figure 3 caption. In the case of the section Materials and Methods, it follows the journal scheme as we already explained.
Comment 19. Line 177-181: also here must enrich the PCA results presentation
Response 19. Thank you for the suggestion. We add one more sentence to enrich the PCA results presentation.
Comment 20. Discussion part: generally good but need to add more references. Some whole paragraphs have no scientific justification.
Response 20. Thank you for the suggestion, we added the references when we think they are essential.
Comment 21. Line 408: department of what? University/ministry?
Response 21. Thank you for this suggestion. The University of Gdańsk in the Faculty of Biology. The sentence was modified to clarify.
Comment 22. Line 426-435: you need to explain what is corresponding to each abbreviation you use.
Response 22. Thank you for this suggestion. Done.
Comment 23. Line 463: the equ 1 should be following the mathematical formula
Response 23. We would like to clarify that the equation was prepared using the Equation function in Microsoft Word. As such, we are unsure what specific improvement Reviewer 2 is requesting.
Comment 24. Line 464: the definitions of the equation symbols should be more carefully defined
Response 24. We appreciate this suggestion. In response, we have updated the manuscript to include more precise definitions of equation symbols. Specifically, we clarified the meanings of Rsample and Rstandard, which are now explicitly defined in the text.
Comment 25. Line 479-481: “ Additionally, Principal Component Analysis (PCA) was used to graphically represent relationships between environmental variables and the δ13C and δ15N values of the 480 studied species.” This part needs to be modified. You should briefly explain what is PCA , based on scientific references and afterwards to say why is PCA useful for your data analysis. Also, some studies using PCA and Macrophyte/similar water quality parameters should be added in the introduction or discussion part and briefly analyze their findings.
Response 25. Thank you for your comment regarding the description of the Principal Component Analysis (PCA) in lines 479–481 and its contextualization in the manuscript. We would like to respectfully note that PCA is a widely known and fundamental multivariate statistical technique for dimensionality reduction and for visualizing patterns in multivariate data. The method has been standard practice for decades (e.g., Jolliffe & Cadima, 2016, Principal Component Analysis: A Review and Recent Developments, Philosophical Transactions of the Royal Society A). Therefore, we believe that providing a detailed explanation of how PCA works would not add value for the intended readership, who can be reasonably expected to be familiar with this method. However, we add the reference mentioned above and we mention in the Materials and Methods section our assumption of PCA design in the presented study.
Round 2
Reviewer 2 Report
Comments and Suggestions for Authors
Thank you for taking my suggestions into consideration. The revised version of the manuscript is significantly improved and suitable for publication. I only have a very minor comment regarding equation (1), new version line 720: please place the equ. (1) text in the right part of the associated mathematical formula.
Author Response
Comment 1
Thank you for taking my suggestions into consideration. The revised version of the manuscript is significantly improved and suitable for publication. I only have a very minor comment regarding equation (1), new version line 720: please place the equ. (1) text in the right part of the associated mathematical formula.
Response 1. Thank you for appreciating our previous improvements. The suggestion about Eq. (1) has been implemented accordingly.